# Static, Dynamic, and Signal-to-Noise Analysis of a Solid-State Magnetoelectric (Me) Sensor with a Spice-Based Circuit Simulator

**DOI:** 10.3390/s22155514

**Published:** 2022-07-24

**Authors:** Yuri Sindler, Simon Lineykin

**Affiliations:** Department of Mechanical Engineering and Mechatronics, Ariel University, Ariel 4077625, Israel; yuri.sindler@msmail.ariel.ac.il

**Keywords:** magnetoelectric (ME) sensors, SPICE, noise analysis, piezoelectric, equivalent circuit modeling, equivalent input noise minimization

## Abstract

Modeling the non-electrical processes by equivalent electrical circuits is a widely known and successfully used technique in research and development. Although finite element methods software development has supplanted electrical analogy techniques due to greater accuracy and intuitiveness in recent decades, the modeling of physical processes based on analogies has several advantages in some cases. Representation of physical processes in the form of lumped circuits and graphs allows researchers to estimate the system with an alternative view, use standardized methods for solving electrical circuits for non-electrical systems, and, most importantly, allows us to use electrical circuit simulators with their unique capabilities. Of particular interest for using the analogy technique are systems that include electronic components along with components belonging to other physical domains, such as mechanical, thermal, magnetic, and others. A solid-state magnetoelectric (ME) sensor equipped with a charge amplifier is proposed in this study as an example of analysis using the equivalent electrical circuit and simulating these circuits using SPICE-based circuit simulators. Sensor analysis is conducted with an emphasis on noise budgeting and optimizing the sensor’s signal-to-noise ratio and resolution. In addition, the steady state, the phasor, and transient types of analyses were employed to study the static and dynamic behavior of the system. Validation of the model using analytical calculations and comparison with experimental data demonstrated superior results.

## 1. Introduction

The magnetoelectric (ME) sensor occupies a niche among the many magnetic field sensors. Its peculiarities are that this type of sensor is passive, solid-state, has no windings, is narrowly directed, has a pico-tesla resolution, and is potentially not expensive [1]. The sensor consists of magnetostrictive and piezoelectric materials with an intermediate material (usually epoxy resin glue). The most used magnetostrictive materials are Terfenol-D and Metglas [2,3,4,5,6]. The most popular piezoelectric ceramic materials include $Pb\;\left(Zr_{x}Ti_{1-x}\;\right)O_{3}\;$(better known as PZT), $Pb\left(Mg_{1/3}Nb_{2/3}\right)O_{3}PbTiO_{3}$ (PMN-PT), and more [6,7]. The ME sensor concept is that a magnetostrictive material reacts to an external magnetic field by generating mechanical stresses. These mechanical stresses are transferred from the magnetostrictive component of the sensor to the piezoelectric component through direct mechanical contact (usually an adhesive bond). The piezoelectric component converts mechanical stresses into electrical charge at its output electrodes. The charge amplifier can measure this signal.

In recent decades, several groups of researchers have carried out in-depth analytical and experimental studies of the properties of ME sensors [6]. They indicated promising materials [8,9], basic mechanical topologies [10,11,12], bias field influence [13], preferred amplifier types [9], and the effect of these factors on sensitivity and sensor resolution [14]. Among the curious facts about ME sensors, we can note that a conventional ceramic capacitor can also operate as an ME sensor because it includes nickel electrodes with magnetostrictive properties and ceramic isolation with a slight piezoelectric effect [15].

Particular attention in the above studies of ME-sensors is paid to analyzing the sensor resolution, signal-to-noise ratio (SNR), and obtaining the equivalent magnetic field noise, particularly at low frequencies. For this purpose, it is necessary to identify all noise sources of the system, specify the noise frequency density of each of these sources, determine the transfer functions between each noise source and the output signal, consider the noise budget, and obtain the spectral density of the resulting noise. Finally, the resulting noise would be divided by the transfer function between the input (magnetic flux density, T) and the output (output voltage, V) to obtain an equivalent magnetic field noise.

Finite element methods (FEM) are often used to analyze internal multiphysics processes in solid elements [2,4]. Efficient software tools for this type of analysis have been developed. However, FEM is difficult to implement to solve the specific problem because not all tools allow mutual simulation of a solid-state component with electronics of various levels of complexity, spectral analysis of signals, noise spectrum density analysis, and the dynamic behavior of a system that includes both solid-state and lumped electronic elements [3]. At the same time, simulation tools, such as Simulation Programs with Integrated Circuit Emphasis (or SPICE-based electronic circuit simulators), can quickly solve such problems, but the solid-state component must be represented as an equivalent electrical circuit. The models based on the Bond graphs have similar capabilities, but the software for such models (for example, MODELICA [16,17]) is still evolving.

The use of electrical analogies for modeling non-electrical processes is a widespread practice. An example would be thermal processes in thermoelectric systems [18], dynamic mechanical processes [19], gas-discharge bulbs behavior [20], subsea observation networks [21,22], and even macro-processes, such as photovoltaic cells and panels [23], urban train traffic [24], and more. This technique is especially effective in cases where the model under study is part of a complex system and must be analyzed and simulated together with the system’s electronic (or electrical) components. The system concerned in this study consists of an amplifier, a feedback network, and the solid-state ME sensor, where magneto-mechano-electrical processes are modeled as an equivalent electrical circuit. In this scheme, all noise sources (intrinsic sensor noise, equivalent noise of the amplifier, noise of feedback elements) that affect the sensor’s resolution are modeled as independent voltage or current sources of white or colored noise without correlation [1,13,25,26,27]. The calculation of the output noise and the equivalent input noise is carried out based on the obtained scheme in an analytical way.

Most SPICE-based circuit simulators allow steady-state, time-domain (transient), and frequency domain (AC) simulation types. Additionally, these simulators allow circuit noise analysis in the frequency domain [28]. This feature is an advantage of the SPICE-based simulators for this specific application over other multiphysics simulation software for systems with lumped parameters, such as MODELICA [16,17]. Moreover, the resistors and semiconductor device models include different white (thermal) and combined (flicker + shot) noise sources ready for noise spectrum density analysis. Therefore, the circuit simulators are intuitively the most suitable tools for analyzing systems that include an electrical part and a non-electrical part, which is expressed as an equivalent electrical circuit, such as ME sensors with amplifiers. However, a few limitations prevent using a circuit simulator for analyzing the proposed system. For example, no element includes the electric charge noise model, the frequency-dependent equivalent series resistance of capacitance of piezoceramics (ESR) noise model, and more.

This work aims to adapt the ME sensor model to be analyzed and optimized using a SPICE-based simulator. For this purpose, sophisticated models of some elements were developed, and universal sources of uncorrelated white and colored noise are presented as separate components included in a circuit consisting of noiseless elements. Simulation of an adapted system model allows for considering parameters usually neglected in analytical calculations, such as a non-ideality of the amplifier, and the frequency dependence of ESR, etc. In addition, we tried to avoid using the Laplace function, which allows for describing the frequency-phase behavior of elements since, outside the frequency domain, such functions slow down simulations and lead to significant calculation errors, which look like a noise in the time domain analysis.

The study [29] was taken as a reference for model validation. The paper analyzes sensor resolution and output signal value when the system is excited by a sine-form input magnetic flux density with an amplitude of 10 nT. The authors obtain essential parameters from the manufacturer’s data and the geometry of materials and compare their analytical calculations with experimentally obtained data. In the current study, we adapted the sensor model presented in the reference study for simulation using a PSPICE-based simulator, performed the simulations using the LTSPICE software [30], and compared the results with our analytical results and the measurement results proposed by the authors of the reference study.

It is important to note that the piezoelectric element is also susceptible to acoustic, thermal, and other noise and interference that complicate measurements. Placing the sensor in vacuum isolation can minimize this additional noise and interference.

The rest of the article is structured as follows: Section 2 provides a brief theoretical background for magnetoelectric sensors, explains the principal coefficients and dependencies, and describes the amplifier topology. The methodology used in the study for an analytical approach to a noise budget calculation and creating the SPICE-oriented equivalent circuits of specialized elements, is demonstrated in Section 3. Noise budgeting using analytical methods and noise analysis in the LTSPICE simulating software and comparison of results are shown in Section 4. Section 5 demonstrates the comparison of the results with experimental measurements. Finally, Section 6 brings us to discussions and conclusions.

## 2. Theoretical Background

In general terms, the effect of magnetostriction can be defined as body deformation in reaction to a change in its magnetization due to exposure to a magnetic field. The effect was first identified in 1842 by James Joule. In [31], the phenomenon of magnetostriction is described, as shown in Figure 1. A magnetic field H [Oe], induced in a magnetostrictive material of length L, by a current-carrying solenoid, as shown in Figure 1a, leads to a change in the geometric size of the material by ΔL along the axis of the field (shown in the figure by a dashed line). The solid thick curve in Figure 1b demonstrates quiescent relative elongation as a function of the applied field. The λ=Δ L/L is independent of the direction of the field, but only on its absolute value and changes from zero to $\lambda_{s}$ value, at which saturation occurs, and the relative elongation no longer depends on the field strength. Minor deviations of the field H about some quiescent value of the field lead to small variations in a λ. This phenomenon can be seen in Figure 1b, depicted by tiny lines and enveloped by dashed lines. A bias field is typically induced using permanent magnets to maximize the ratio of small λ and H signals [32]. The value of $H_{bias}$ is often determined empirically. Knowing the λ and the coefficient of elastic deformation, it is possible to calculate the mechanical stress in the material under the influence of a magnetic field, both quiescent and small signal.

By its definition, the piezoelectric material can accumulate an electric charge in response to mechanical stress. This property of the piezoelectric material is reversible. That is, the piezoelectric material demonstrates mechanical deformations in response to an electric field. However, we will consider only the first, direct relationship in this study.

Figure 2a shows one of many possible magnetoelectric sensor topologies. This topology is published in [29], and we use this publication to validate the proposed model. This topology includes a PMN-PT piezo-fiber element (compiled in an optimized way) sandwiched between two plates stacked with six thin layers of Metglas magnetostrictive material. The layers are bonded mechanically with epoxy resin. The choice and optimization of the topology are described in detail in [29]. The lead magnesium niobate–lead titanate (PMN-PT) single crystals exhibit ultrahigh piezoelectric coefficients of approximately $2000\;pC\cdot N^{-1}$ and a low tan δ value of roughly 0.005. This material is an epoxy matrix of piezoelectric elements with embedded electrodes. The geometry of the interdigitated (ID) electrodes is such that the composite is configured in a multi-push-pull modality. The reader can find more information about such materials and methods for their modeling in [33]. The sensor is equipped with permanent magnets to create a bias field. Permanent magnets are fixed on both sides of the sensor at a certain distance, providing the optimal value of the bias field. Both biasing and magnetic fields are directed along the sensor. The ME-sensor can operate at both high (ultrasonic) resonant and low and quasi-static frequencies. However, in the field of view of this study is a sensor capable of capturing a signal at low and ultra-low frequencies. At these frequencies, problems arise associated with the intrinsic noise of the piezoelectric element and the noise of amplifying devices.

The block diagram in Figure 2b shows the chain of converting the energy of a magnetic signal first into mechanical stress using a magnetostrictive component and then into an output electrical signal. The resulting electrical signal can be measured as an electrical charge using a charge amplifier or as a voltage across the output capacitance of a piezoelectric component.

The magnetoelectric sensor is a multi-domain system that includes magnetic, mechanical, and electrical domains. One of the conventional methods for analyzing such systems is a method of analogies. The idea of the method is that an analogous process in one domain can emulate the original processes from another domain if identical equations describe both. In the case of a magnetoelectric sensor, it is convenient to describe all non-electrical processes using equivalent electrical processes.

In Figure 3a, the equivalent electrical circuit of a magnetoelectric sensor is shown. The electrical circuit includes a controlled current source connected in parallel to the equivalent resistance $R_{loss}$, which describes electrical losses in the piezoelectric material and capacitor $C_{pz}$, which is formed due to the dielectric properties of the piezoceramics between the output electrodes. The dielectric losses in ceramics are represented by the frequency-dependent equivalent series resistance ESR. The current $I_{s}$ of the controlled current source is equal to the time derivative of the electrical charge q, generated by the piezoelectric material in response to mechanical stress. The charge generated by the ME sensor is proportional to the magnetic field H with the coefficient $\alpha_{Q\;}$. The coefficient $\alpha_{Q\;}$, in turn, is shown in the graph Figure 3b as a function of the constant bias field $H_{bias}$. The data for the plot are taken from an article [29] for a specific sensor. A factor of 10 k is needed to match the input signal, the magnetic flux density measured in tesla, and the magnetic field H measured in oersted, with magnetic permeability $\mu_{0}$ corresponding to free space.

The output signal can be measured as a voltage using a voltage amplifier. However, using a trans-impedance amplifier with a capacitive characteristic may be a better alternative due to higher stable gain and the possibility of setting the amplifier’s bandwidth. Such an amplifier is often called a charge amplifier. When such an amplifier topology is used, the output voltage is proportional to the piezoelectric element’s charge, internal mechanical stress, and magnetic input signal. The basic topology of such an amplifier is demonstrated in [13].

Every non-reactive element, such as a resistor or equivalent resistor and an amplifier, generates some amount of white or colored noise. These noises impact the output signal and reduce the sensor’s resolution. Choosing the correct amplifier elements through noise balance optimization maximizes sensor resolution. The modeling technique proposed in this study allows for easy calculation of the ME sensor’s equivalent input magnetic field noise to be performed quickly and more accurately than the simplified analytical calculations demonstrated in the cited papers.

## 3. Methodology

This work’s declared goal is to create an ME sensor model that would allow all kinds of simulations available with SPICE-based tools: transient analysis, DC-analysis, AC analysis, noise analysis, and the like. Furthermore, the model must be compatible with other SPICE elements, particularly with operational amplifier models (specific and generic). Finally, we will demonstrate the model validity by comparing the simulation results with the analytical calculations and laboratory measurements made on an actual sensor. For this, a description of modeling and analysis methods used in the study will be collected and demonstrated below.

### 3.1. Analytical Approach to a Noise Budget Calculation

Mathematical analysis is a widely accepted method for noise budget calculation, noise minimization, and sensor resolution improvement. Detailed mathematical analysis of such a system may be found in [14,29]. The authors propose a detailed analytical model of noise sources in a piezoelectric component of the ME sensor and charge amplifier, a feedback circuitry noise, and an intrinsic noise of an operational amplifier. A common assumption is that the impact of noise in magnetostrictive components is minor relative to the noise of the piezoelectric component. Following [1,14], where the authors calculate an equivalent magnetic noise floor, we represent the total noise of the system in the form of equivalent magnetic noise. Unlike the mentioned research where all noise sources are represented as equivalent charge sources, in the present study all noise sources are represented as equivalent voltage and equivalent current sources. This method allows us to further use the noise sources developed in the PSPICE simulator.

Analytical calculation of the noise budget includes defining an input-to-output transfer function that relates the input magnetic signal to the output voltage and an array of transfer functions that relate each of the lumped noise sources and the output signal. For this purpose, the simplified methodology based on that proposed in [34,35] is used. The simplification comes down to the fact that at low frequencies, the output impedance of the operational amplifier is negligible compared to the feedback impedance, which means that the feed-forward component can be neglected, and the block diagram of the feedback amplifier looks like the one shown in Figure 4a.

The value of G can be extracted by disabling the error signal $s_{\varepsilon }$ as shown in Figure 4b. In this case:(1)$$G\equiv \frac{s_{\varepsilon }^{\prime }}{s_{s}},$$
where $s_{\varepsilon }^{'}$ is the output signal of the summing unit under $s_{\varepsilon }$ that is set to zero.

For extracting the β value, the circuit with the disabled source signal $s_{s}$ and nonzero output signal $s_{o}$ shown in Figure 4c can be used. In this case, an input signal is set to zero. Thus:(2)$$\beta \equiv -\frac{s_{\varepsilon }^{\prime \prime }}{s_{o}}\;\;,$$
where $s_{\varepsilon }^{\prime \prime }$ is the output of the summing unit under the zero source signal condition.

Thus, when $G_{s}\left(s\right)$ and $\beta_{s}\left(s\right)$ are known, the transfer function connecting the source signal and the output signal looks as follows:(3)$$H_{s}\left(s\right)={\frac{G_{s}\left(s\right)\cdot A_{ol}\left(s\right)}{1+\beta_{s}\left(s\right)\cdot A_{ol}\left(s\right)}|}_{\beta \left(s\right)\cdot A_{ol}\left(s\right)\gg 1}\approx \frac{G_{s}\left(s\right)}{\beta_{s}\left(s\right)}$$

The subscript s means that the blocks G and the β correspond to a source, and variable s is a Laplace variable. The simplification proposed in Expression (3) is valid only if $\beta \left(s\right)\cdot A_{ol\left(s\right)}\gg 1$.

As an illustration of the application of the proposed method, we propose to consider the ME sensor circuit equivalent circuit together with the charge amplifier proposed in [14,26,27] and generally used with piezoelectric sensors [3], as shown in Figure 5. The results of this example will be used in further calculations.

To derive a transfer function connecting the source signal B to the output voltage $H_{B}\left(s\right)$ following the method presented in Figure 5, one should extract the expressions for $G_{B}\left(s\right)$ and $\beta_{B}\left(s\right)$ by splitting the original circuit into two simplified ones, if necessary, as depicted in Figure 6. Setting the original error signal $v_{\varepsilon }\;$ to zero causes the opamp output to go to zero, as shown in Figure 6a. In this case, the auxiliary error signal $v_{\varepsilon }^{'}$ can be derived as:(4)$$v_{\varepsilon }^{\prime }=-\left(B\cdot 10k\cdot \alpha_{Q}\cdot s\right)\cdot R_{loss}∥\left(\frac{1}{sC_{pz}}+ESR\right)∥R_{in}∥R_{f}∥\frac{1}{sC_{f}}$$

Thus:(5)$$G_{B}\left(s\right)\equiv \frac{v_{\varepsilon }^{\prime }}{B}=-\frac{10k\;\alpha_{Q}R_{f}\;\left(1+ESF\;C_{pz}\;s\right)s}{1+R_{f}\left(\frac{1}{R_{in}}+\frac{1}{R_{loss}}\right)+\left(ESR\;C_{pz}\left(1+R_{f}\left(\frac{1}{R_{in}}+\frac{1}{R_{loss}}\right)\right)+\left(C_{pz}+C_{f}\right)R_{f}\right)s+\left(C_{f}C_{pz}ESR\;R_{f}\right)s^{2}}$$

Setting the source signal B to zero causes the second simplified circuit shown in Figure 6b. The error signal for this simplified circuit $v_{\varepsilon }^{\prime \prime }$ can be derived in this case as:(6)$$v_{\varepsilon }^{\prime \prime }=-v_{out}\cdot \frac{R_{loss}∥\left(ESR+\frac{1}{s\;C_{pz}}\right)∥R_{in}}{R_{loss}∥\left(ESR+\frac{1}{s\;C_{pz}}\right)∥R_{in}+R_{f}∥\frac{1}{s\;C_{f}}}\;,$$
and:(7)$$\beta_{B}\left(s\right)\equiv -\frac{v_{\varepsilon }^{\prime \prime }}{v_{out}}=\frac{\;\left(1+ESF\;C_{pz}\;s\right)\left(1+C_{f}R_{f}\right)}{1+R_{f}\left(\frac{1}{R_{in}}+\frac{1}{R_{loss}}\right)+\left(ESR\;C_{pz}\left(1+R_{f}\left(\frac{1}{R_{in}}+\frac{1}{R_{loss}}\right)\right)+\left(C_{pz}+C_{f}\right)R_{f}\right)s+\left(C_{f}C_{pz}ESR\;R_{f}\right)s^{2}}$$

Therefore, the transfer function relating the magnetic signal and the amplifier’s output voltage is as follows:(8)$$H_{B}\left(s\right)=\frac{A_{ol}\left(s\right)G_{B}\left(s\right)}{1+\beta_{B}\left(s\right)A_{ol}\left(s\right)}\approx \frac{G_{B}\left(s\right)}{\beta_{B}\left(s\right)}=-\frac{10k\;\alpha_{Q}R_{f}\;s}{1+C_{f}R_{f}\;s}\;.$$

Expression (8) shows that the resulting simplified transfer function of the system is equivalent to that which could be derived using the “virtual zero” concept. Nevertheless, the proposed method makes it possible to consider the poles and zeros of the transfer function of the operational amplifier and conclude the system’s stability. An additional convenience of this approach is the ease of noise optimization in the presence of multiple noise sources, assessing their impact on the output voltage signal, and deriving the equivalent magnetic noise of each of the sources.

### 3.2. SPICE-Oriented Equivalent Circuits for Specialized Elements

The following methodological technique used in this study is the compiling of equivalent circuits emulating non-electrical processes, such as differentiation of charge over time, the dependence of $\alpha_{Q}$ on the bias magnetic field, making controlled white and colored noise sources utilizing the built-in PSPICE capabilities. Examples of such equivalent circuits are demonstrated in subsequent sub-sections.

#### 3.2.1. Time Derivative and Time Integral as an Electrical Circuit Element

The SPICE-based simulation tools allow us to use time integration, time derivative, and more complex transfer functions in the Laplace expressions or frequency-gain-phase tables in behavioral sources. Such a description of the dynamic system is especially effective for phasor (AC) simulations and noise analysis in the frequency domain. Nevertheless, the expressions in such forms in the time domain lead to significant “calculation noise” due to the inverse Laplace transform. This disadvantage is significant when simulating systems with low noise levels. Based on the above, we propose “natural-like” integrating and differentiating electrical circuits equally suitable in both the time and frequency domain simulations. Thus, Figure 7 depicts examples of the corresponding circuits. The current $i_{c}$ is the time derivative of the source voltage $v_{c}$ in Figure 7a, and the potential drop over the capacitor $v_{c}$ is equal to the time integral of the current source $i_{c}$ in Figure 7b as follows from the current to voltage and voltage to current relations of the capacitor:(9)$$i_{c}\left(t\right)=C\frac{dv_{c}\left(t\right)}{dt}\;$$
and:(10)$$v_{c}\left(t\right)=\frac{1}{C}\int^{\;}i_{c}\left(t\right)dt$$
where $v_{c}\;\left(t\right)$ and $i_{c}\;\left(t\right)$ are the voltage and the current of the capacitor, respectively.

#### 3.2.2. White Noise and 1/f Universal Sources

Some components, such as resistors, diodes, transistors, and more, contain built-in noise sources that make noise simulation in the frequency domain available. However, it is sometimes necessary to build models and controlled noise sources using parameters obtained from manufacturer data, analytical calculations, or experimental data. For this purpose, an equivalent universal controllable noise source, modified from the one proposed in [36], is proposed.

The standard diode model includes a noise generator model. Shot noise and flicker noise generated by the DC current $I_{D}$ through the diode are characterized by the following spectral density:(11)$$\frac{\left\langle i_{d}^{2}\right\rangle }{\Delta f}=\underset{shot\;noise}{\underset{︸}{2\;q\;I_{D}}}+\underset{flicker\;noise}{\underset{︸}{k_{f}\frac{I_{D}^{a_{f}}}{f^{f_{fe}}}}}\;,$$
where $i_{d}$ is a current noise of a diode, ·f is the noise bandwidth, $I_{D}$ is a DC current of the diode, q is the electron charge, $k_{f}$ is a flicker noise coefficient, $a_{f}$ is a flicker noise exponent, $f_{fe}$ is a flicker noise frequency exponent, and f is the simulation frequency. By default, the values of the flicker noise coefficients are $\;k_{f}=0$, $a_{f}=1$, and $f_{fe}=1$; that is, only the shot noise (white) component is generated, and the flicker noise component is neglected. The nonzero value of the $k_{f}$ means the flicker noise of the diode is also considered within the simulation. The current noise knee frequency $f_{ci}$ where the flicker noise value is equal to the shot noise value can be specified using the $k_{f}$. Thus, at the knee frequency $f_{c}$:(12)$$2qI_{D}=\frac{k_{f\;}I_{D}}{f_{ci}}\;,$$
thus:(13)$$K_{f}=2\;q\;f_{ci}\approx 3.2\cdot 10^{-19}\;f_{ci}\;$$

An example of a white and flicker noise generator is shown in Figure 8a. The diode model $D_{1}$ is called Dd and differs from the default diode model by setting a nonzero flicker noise coefficient $k_{f}$ according to Expression (13). The frequency $f_{ci}$ is set as a global parameter using the **param** directive. The constant current source $I_{b}$ sets the bias current of diode $D_{1}$ to 1 A. The shot noise of the diode under these conditions is $560\mathrm{pA}/\sqrt{\mathrm{Hz}}$. The capacitor $C_{1}$, which has a huge capacitance of $1\cdot 10^{9}\;F$, splits the diode’s current into a DC component flowing through the diode $D_{1}$ and a noise component flowing through the capacitor $C_{1}$. As a result, the capacitor’s current (the current noise) can be measured using a zero-voltage source $V_{p}$ as current i(Vp). The behavioral voltage source $b_{1}$ generates a noise voltage at a node **out**. A factor of $1.77\cdot \;10^{9}$ makes the shot noise value equal $1\;V/\sqrt{\mathrm{Hz}}$. Thus, the coefficient $e_{n}$, that is the value of the desired white noise spectrum density, which is set as a parameter, makes the spectral density of the resulting white noise equal to $e_{n}$. A phasor source (AC) $V_{1}$ with an amplitude of 1 V is needed as a source for noise analysis. The directive **noise** establishes noise analysis in the frequency range from 10 mHz to 1 kHz with 101 points per decade resolution. The spectral density of the noise generated by the circuit is shown in Figure 8b. The graph shows that the noise becomes “white” at high frequencies, and its density is $e_{n}\approx 10\;\mathrm{pV}\sqrt[{-1}]{\mathrm{Hz}}$. At a frequency of 1 Hz, the noise density in $\sqrt{2}$ is higher than $e_{n}$, and over the frequency range from 10 mHz to 100 mHz, the noise density drops by a factor of $\approx \sqrt{10}$.

#### 3.2.3. ESR Equivalent Circuit

The ability of dielectric materials to dissipate potential energy in the form of heat is typically expressed using a dissipation factor (DF), also known as tan(δ). In the literature, a circuit including an ideal capacitor and an equivalent series resistor (ESR) can, to a first approximation, emulate the presence of dielectric losses. ESR depends on the properties of the dielectric material, such as bulk dielectric conductivity σ, dielectric constant ε, lossless capacitance C, and ω, an AC frequency of current i [37]:(14)$$ESR=\frac{\sigma }{\varepsilon \omega^{2}C}=\frac{\tan\left(\delta \right)}{\omega C}\;,$$
where:(15)$$\tan\left(\delta \right)=DF=\frac{\sigma }{\varepsilon \omega }$$

Manufacturers of dielectric materials and piezoelectric ceramics that are dielectric materials typically provide tan delta values corresponding to a frequency of 1 kHz. Around this frequency, the ESR value can be considered constant and frequency independent. Such a model is visual and easy for calculations but challenging to implement in a simulator in the form shown in Expression (14). Thus, if tan(δ) has a constant value, the ESR’s absolute value decreases with frequency (like impedance of a capacitor), but its phase shift remains zero. There is no element in PSPICE with such properties. Several known methods for building an equivalent electrical circuit of the ESR, such as the Debye model or the Cole–Cole model, are based on the material’s properties [38,39], including multiple capacitive and resistive elements. In this study, however, we will use a simplified method, following the authors of the reference articles [13,29]. Two assumptions will be made for this purpose: a. The tan(δ) is constant and b. The resistive value of the ESR does not affect the system’s transfer function because it is much less than the series capacitor impedance. However, the thermal noise generated by the ESR must be considered. The following expression can express the voltage noise generated by ESR:(16)$$\frac{e_{ESR}^{2}}{\Delta f}=4\cdot k_{b}\cdot T\cdot \underset{ESR}{\underset{︸}{\frac{\tan\left(\delta \right)}{2\;\pi \;f\;C_{pz}}}}=\frac{1}{f}\cdot \frac{2\cdot k_{b}\cdot T\cdot \tan\left(\delta \right)}{\pi \;C_{pz}}$$
where $k_{b}$ is Boltzmann’s constant, T is operating temperature in kelvin, and Δf is the noise bandwidth.

Thus, such noise can be modeled using a flicker noise model depicted in Figure 8a and expressed in (11) with the corner frequency outward. Let us set the corner frequency as $f_{c}{}_{ESR}=1\mathrm{kHz}$. In this case, the white noise spectrum density will be equal to:(17)$$e_{n_{ESR}}=\sqrt{\frac{1}{f_{C_{ESR}}}\cdot \frac{2\cdot k_{b}\cdot T\cdot \tan\left(\delta \right)}{\pi \;C_{pz}}}$$
with the equivalent circuit implementation of the ESR together with the ideal capacitor shown in Figure 9.

#### 3.2.4. Emulation of the Dependence of the Coefficient $\alpha_{Q}$ on the Bias Field

The coefficient $\alpha_{Q}$, part of the diagram of Figure 3a, depends on an external constant magnetic bias field. The relationship between the $\alpha_{Q}$ and $H_{bias}$ is shown graphically in Figure 3b [29]. Since the curve for $\alpha_{Q}$ is derived from the curve shown in Figure 1b multiplied by the magnetostrictive and the piezoelectric constants of the corresponding materials, we can restore the quiescent dependence of the charge on the magnetic field by integrating the $\alpha_{Q}$ function over $H_{bias}$. The resulting function can be entered into the model as a table. Figure 10a demonstrates a hierarchical block, including the left side (from input signal B to charge signal q) of the equivalent circuit shown in Figure 3a. This block has two inputs: one for the input signal B (T) and the second for the bias field HbiasOe (Oe). The input signal of the magnetic field HinOe (Oe) corresponds to the magnetic flux density Bin (T) in free space. The input signal HinOe is added to the bias signal HbiasOe by means of behavioral source B1, and the result is multiplied by the table function plotted in Figure 10b.

## 4. Noise Budget (Analytical Calculation)

Before proceeding with the validation of the proposed model, we propose to carry out an analysis of the noise of the ME-sensor system together with an amplifier (see the topology of Figure 5), as conducted in [13], to select the optimal operational amplifier and optimize the feedback element connections. For this purpose, one should distinguish each noise source in the topology, derive its transfer function to the output, and finally calculate the equivalent input noise by dividing the total output noise by the input-to-output transfer function shown earlier in the Expression (8).

All noise sources considered in this work are mapped in Figure 11. Among them, the thermal noise current source $i_{loss}$ of the resistor $R_{loss}$, the voltage noise source $e_{ESR}$ corresponding to the ESR noise, the opamp’s equivalent current, and voltage noise sources $i_{n}$ and $e_{n}$ correspondently, and $i_{f}$ the current noise source of the feedback resistor $R_{f}$. For each of the noise sources, the transfer function can be expressed using the $G_{x}$ and $\beta_{x}$ blocks as shown above, (1), (2), where the subscript x indicates the noise source. The considered noise source x must be taken as nonzero, with all the other sources set to zero. It is not difficult to show that the block $\beta_{x}$ will be the same for all transfer functions and equal to $\beta_{B}$, see (7). Thus, only $G_{x}$ blocks need to be derived for each noise source.

All the $G_{x}\left(s\right)$ blocks are tabulated in Table 1. When compiling the table, the following simplifications were made: the ESR resistance is assumed as small relative to the impedance of the $C_{pz}$ and does not affect the transfer functions. The $R_{in}$ resistor of the selected opamp is large enough (about 10T Ω) relative to $R_{loss}$ and can also be neglected for simplicity.

Having the data in Table 1, it is possible to express analytically the noise spectral density at the output of the amplifier, $onoise,\;V\cdot Hz^{-\frac{1}{2}}$:(18)$$onoise\approx \frac{1}{\beta_{B}\left(s\right)}\;\sqrt{\left(i_{loss}^{2}+i_{n}^{2}+i_{f}^{2}\right)\cdot G_{lnf}^{2}\left(s\right)+e_{ESR}^{2}\cdot G_{esr}^{2}\left(s\right)+e_{n}^{2}\cdot G_{n}^{2}\left(s\right)}\;.$$

In addition, the equivalent spectral density of the input signal, $inoise,\;T\cdot Hz^{-\left(1/2\right)}$, which enables supposing the resolution of the sensor can be derived as:(19)$$inoise\approx \frac{\beta_{B}\left(s\right)\cdot onoise}{G_{B}\left(s\right)}=\sqrt{\left(i_{loss}^{2}+i_{n}^{2}+i_{f}^{2}\right)\cdot \frac{G_{lnf}^{2}\left(s\right)}{G_{B}^{2}\left(s\right)}+e_{ESR}^{2}\cdot \frac{G_{esr}^{2}\left(s\right)}{G_{B}^{2}\left(s\right)}+e_{n}^{2}\cdot \frac{G_{n}^{2}\left(s\right)}{G_{B}^{2}\left(s\right)}}$$

Thus, the designer can identify the dominant noise source among those inherent in the sensor ($i_{loss},\;e_{ESR}$) and choose the opamp and feedback resistor $R_{f}$ in such a way as to minimize the influence of corresponding noise sources $i_{n}$, $e_{n}$, and $i_{f}$:(20)$$i_{f}\leq i_{loss}\;\;\mathrm{and}\;\;i_{f}\leq e_{ESR}\cdot \frac{G_{ESR}\left(s\right)}{G_{lnf}\left(s\right)}\;,$$
(21)$$i_{n}\leq i_{loss}\;\;\mathrm{and}\;\;i_{n}\leq e_{ESR}\cdot \frac{G_{ESR}\left(s\right)}{G_{lnf}\left(s\right)}$$
(22)$$e_{n}\leq i_{loss}\cdot \frac{G_{lnf}\left(s\right)}{G_{n}\left(s\right)}\;\;\mathrm{and}\;e_{n}\leq e_{n}\cdot \frac{G_{ESR}\left(s\right)}{G_{n}\left(s\right)}$$

Or, for topology with a charge pump amplifier, shown in Figure 11:(23)$$R_{f}\geq R_{loss}\;\;\;\;\mathrm{and}\;\;\;R_{f}\geq 2\pi fC_{pz}\tan\left(\delta \right)\;,$$
and:(24)$$i_{n}\leq 2\sqrt{R_{loss}k_{b}T}\;\mathrm{and}\;i_{n}\leq 2\sqrt{2\;\pi \;f\;C_{pz}\tan\left(\delta \right)k_{b}T}\;.$$

The $i_{f}$, the thermal noise of the feedback resistor $R_{f}$, and the opamp equivalent input noise sources $i_{n}$ and $e_{n}$ contribute to the equivalent field noise to the same extent as the noises of $R_{loss}$ and ESR in the case of equality in Expressions (23)–(25) and to a lesser extent in the case of inequality. If the values on both sides of the inequality differ by a factor of three or higher, then the influence of such a noise source is negligible compared to the dominant noise source. In the sensor example discussed in [29], at frequencies below 1 Hz, the equivalent resistor noise $R_{loss}$ is the dominant noise source of the sensor. The ESR noise becomes the dominant noise of the sensor at higher frequencies.

## 5. Validation of Simulation Results vs. Experimental Measurements

The experimental data published in [29] will be used to validate the proposed model in this study. The parameters of the ME sensor used in the reference article are summarized in Table 2. The data collected in this table are obtained from laboratory measurements of the prototype. The table also shows the value $\alpha_{Q}$ that corresponds to the maximum value of the plot $\alpha_{Q}$ vs. $H_{bias}$ shown in Figure 3b. In addition to the data of the sensor, the gain and minimum bandwidth of the charge amplifier used in [29] are given in the table. The authors of [29] do not specify either the topology or the amplifier’s circuitry. However, another article [14] written by the same researchers proposed a basic amplifier topology, shown in Figure 5, that we used as a template.

Minimizing the noise using Expressions (23)–(25) is necessary to build an amplifier. The noise requirements for the opamp and feedback resistor consider that their noise sources contribute to the equivalent input magnetic field noise to the same extent as the sensor’s internal noise sources. These values are different for different frequencies and are tabulated in Table 3.

The requirements specified in the table must be narrowed if the noise of the amplifying circuitry is required to be negligible compared to the intrinsic noise of the sensor. The feedback capacitor $C_{f}$ determines the gain of the charge amplifier. The feedback resistor $R_{f}$ is placed in parallel with the feedback capacitor to avoid the output voltage rolling out into the non-linear region. As one can see from Table 3, the $R_{f}$ needs to be fairly large so that its impact on the equivalent input noise is negligible.

An operational amplifier with the characteristics indicated in the table is not available in today’s integrated circuits market. However, such an amplifier can be, possibly, explicitly designed for this specific sensor by creating an ultra-low-noise input stage to a standard opamp. Still, the creation of such an amplifier is beyond the scope of this study. So, instead, this work uses the common LMC6044 ultra-low noise amplifier, as in [14]. A second amplification stage must be added to a circuit to obtain the required gain of $5.18\;V\cdot {\mathrm{pC}}^{-1}$. However, the noise characteristics of the second opamp have an insignificant effect on the equivalent input noise, just enough to be low noise. The LTSPICE simulation circuit is shown in Figure 12.

All schematic parameters are summarized in Table 4. All directives in SPICE start with a dot. A semicolon at the beginning of a line disables the directive. Different simulation profiles are used for different types of analysis. The various simulations are run here for model validation. Finally, all the results are compared with the article’s experimental data.

### 5.1. Charge Amplifier’s Gain vs. Bias Magnetic Field

Figure 13a shows the simulation result in the frequency domain. The AC simulation directive is set as active; scanning is performed in decades from 100 mHz to 100 Hz, with a resolution of 1001 points per decade. As shown in Figure 13a, in the frequency range of 0.1 Hz to 100 Hz, the gain of the charge amplifier is $\approx 5.2\;\mathrm{TV}\;C^{-1}=5.2\;V\;{\mathrm{pC}}^{-1}$. This corresponds to the gain of the charge amplifier used by the article’s authors in [29], which we chose as a reference and is given in Table 2.

The authors of the reference article report that with a magnetic bias field of approx. 8 Oe, the output signal of the amplifier demonstrates 1.4 V with an input signal of 10 nT at a frequency of 1 Hz. The graph in Figure 13b was obtained by scanning the hbiasOe parameter (bias magnetic field H in oersteds) using the “.step” directive. This statement allows us to run an AC sweep analysis many times for different values of the sweep parameter. In our case, the analysis was run 124 times for the sweep parameter scanned linearly from 1 mOe to 12.25 Oe with a 100 mOe step. The “.meas” directive extracts the values of the output voltages at a frequency of 1 Hz of each run and builds a table of the output voltage vs. sweep parameter. This table is presented graphically in Figure 13b. The figure shows that the maximum value of the output voltage occurs at a bias field value of 8 Oe. This value is the same as was measured in the paper. The value of the output voltage at this bias-point with an input signal having an amplitude of 10 nT is 1.4 V. That is, precisely the value that was measured in the paper using a lock-in amplifier at a frequency of 1 Hz.

Figure 14 demonstrates simulation results of the same experiment proposed by the authors of the cited article [29] but now in the time domain. The circuit under simulation shown in Figure 12 has a voltage source generating an equivalent magnetic field with a sine waveform with an amplitude of 10 nT and a frequency of 1 Hz at the sensor’s input. The simulation time is set to 200 s for all transients to complete. In Figure 14a, the last two cycles of the output voltage are shown. It can be seen from the figure that the amplitude of the output voltage signal is 1.4 V, which corresponds to the results of the original experiment. In Figure 14b, the dependence of the output voltage amplitude on the biasing magnetic field is demonstrated. The magnitude of the biasing magnetic field is changed using the **param** directive from minimal values up to 12.5 Oe, as in the original experiment in [29]. Measuring the output signal’s amplitude for each step of the scanned parameter was carried out using the **measure** directive.

### 5.2. Equivalent Input Noise Spectral Density

PSPICE is suitable for noise analysis in the frequency domain. The “.noise” directive calculates the noise spectral density at the output terminal (onoise) or the equivalent noise at the input source (inoise). In addition, the algorithm calculates the **gain** as the ratio of the output and input signals. This allows us to analyze the contribution of each noise source to the equivalent input noise. Each noise source’s impact on the output should be divided by the gain for this sake. Figure 15 shows the total equivalent input noise spectral density and its components. One can see from the chart that for the proposed ME sensor with the chosen amplifiers, the dominant noise source is $i_{n}$, the equivalent current noise of the amplifier U1. Therefore, this noise dictates the equivalent input noise level.

Knowing the values of the equivalent current and voltage sources, it is possible to mathematically extract the equivalent input noise of the sensor by taking the root of the difference of the squares of the total measured noise and the total noise of the amplifier. So, from (19), the noise floor of the ME sensor is:(25)$$noisefloor\approx \sqrt{inoise^{2}-i_{n}^{2}\cdot \frac{G_{lnf}^{2}\left(s\right)}{G_{B}^{2}\left(s\right)}-e_{n}^{2}\cdot \frac{G_{n}^{2}\left(s\right)}{G_{B}^{2}\left(s\right)}}\;.$$

The noise floor of an ME sensor can be modeled using the proposed model, as shown in Figure 16a. For this purpose, all amplifier noise sources should be disabled. The noise of the feedback resistor does not need to be nulled since its noise value is negligible compared to the intrinsic noise of the ME sensor. The equivalent magnetic noise and its component simulation results are shown in Figure 16a. The simulated equivalent magnetic noise floor and the equivalent input noise measured and extracted from measurements published in paper [29] are compared in Figure 16b.


## 6. Discussion and Conclusions

The presented model of the ME-sensor in the form of an equivalent electrical circuit makes it possible to simulate the model in SPICE-based electrical circuit simulators together with signal amplification and conditioning circuits of any complexity. The model is compatible with all types of analysis offered by the circuit simulator, such as steady-state (DC), time-domain (transient), phasor (AC), and noise spectral density (noise) analysis types. The model parameters can be extracted from the material manufacturer’s data sheets as well as by experimental means. In addition, the model can be adapted to a wide range of solid-state magnetoelectric sensors and is not limited to specific materials, geometries, and topologies. The model allows for the simulation of serial and parallel as well as mixed connections of an unlimited number of sensors of the same or similar type. The ME sensor characteristics generated by simulations using the SPICE-based LTSPICE software do not contradict but fully support the analytical calculations proposed here and are provided by other research, such as [1,13,14,27,29], and correspond to experimental results published in [14,29].

The model can be extended to obtain greater accuracy and proximity of the results to experimental data. More complex ESR modeling can be performed using a piezoelectric ceramic model with quasi-distributed parameters, such as the Debye model or the Cole–Cole model, are based on the material’s properties [38,39], including multiple capacitive and resistive elements. Emulation of the elastic properties of ceramics using an equivalent inductor will allow simulation of the ME-sensor at frequencies close to resonance. Thermal losses and corresponding noise caused by the plasticity of the adhesive layer and the magnetostrictive components, as well as the influence of their masses on the resonant frequency, can also be added to a model to improve accuracy.

## Figures and Tables

**Figure 1 sensors-22-05514-f001:**
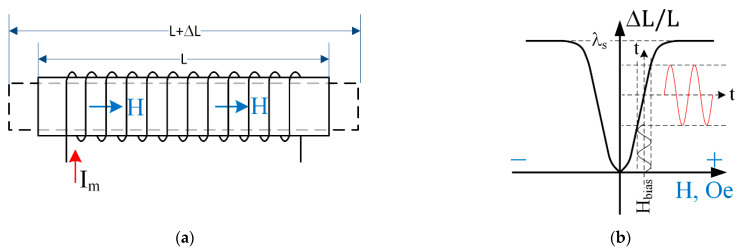
A magnetostrictive material exhibits a response to a magnetic field by variation in its dimensions in the direction of the field. (**a**) The solenoid carries the DC, and the AC electric current induces the field. (**b**) Relative elongation of the material due to the field. The large (quiescent) signal is shown with a thick solid line, and the small field/deformation signals are shown by thick solid lines enveloped by dashed lines.

**Figure 2 sensors-22-05514-f002:**
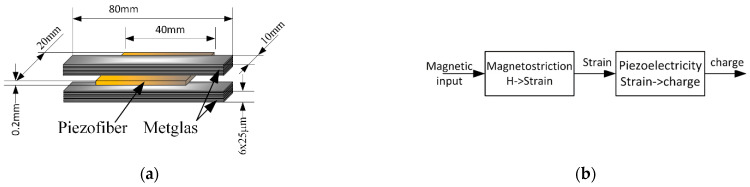
An example of a magnetoelectric sensor was proposed in [29]. (**a**) Mechanical topology and dimensions. All the layers are bonded mechanically with epoxy resin. (**b**) Block diagram of multi-domain signal conversion starting from the magnetic domain to the mechanical domain and then to the electrical domain.

**Figure 3 sensors-22-05514-f003:**
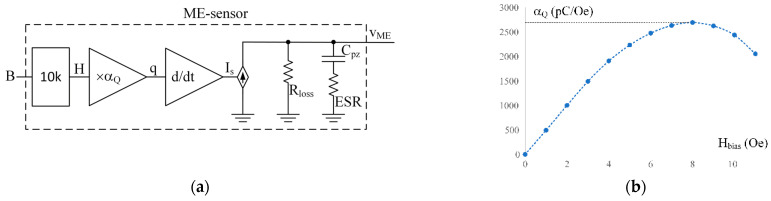
The equivalent diagram of the ME sensor is shown in (**a**). The charge q is proportional to the field H with the coefficient $\alpha_{Q}$. The $\alpha_{Q}$ vs. $H_{bias}$ is shown in (**b**). The data for the plot adapted with permission from ref. [29]. 2011, John Wiley and Sons. An input signal is a magnetic flux B [T] corresponding to a magnetic field H[Oe] in the air with a factor of 10 k.

**Figure 4 sensors-22-05514-f004:**
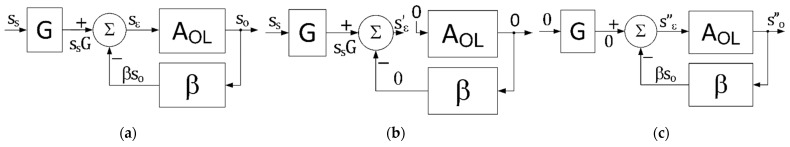
The block diagram of the system with feedback. (**a**) General form, where $A_{ol}$ is an open loop gain of the amplifier, G is a constant gain, β is a feedback gain, and Σ is a summing unit. The signals are: $s_{s}$ is a source signal, $s_{o}$ is an output signal, and $s_{\varepsilon }$ is an error signal. (**b**) The circuit with the disabled error signal $s_{\varepsilon }$ and nonzero source signal $s_{s}$. (**c**) The circuit with the disabled source signal $s_{s}$ and nonzero output signal $s_{o}$.

**Figure 5 sensors-22-05514-f005:**
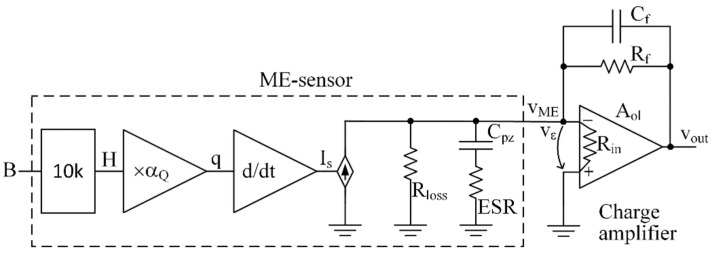
ME-sensor equivalent scheme (Figure 3) with charge amplifier circuitry. The charge amplifier consists of an opamp with open-loop gain $A_{ol}$, input impedance $R_{in}$, and the feedback elements $C_{f}$ and $R_{f}$ that form the gain and bandwidth. An output impedance of the opamp is negligible compared to a feedback resistor at low frequencies.

**Figure 6 sensors-22-05514-f006:**
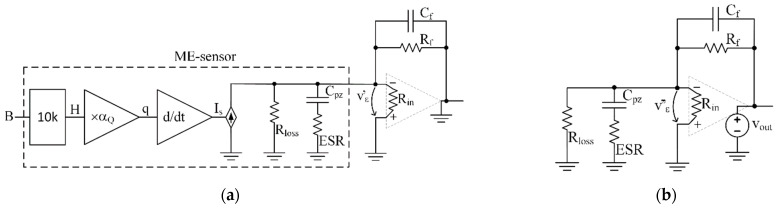
Simplified schemes for defining the components of a block diagram of a feedback system: (**a**) the error signal $v_{\epsilon }^{'}$ is suppressed while the input signal B is nonzero and (**b**) the input signal B is suppressed while the output signal $v_{out}$ is nonzero.

**Figure 7 sensors-22-05514-f007:**
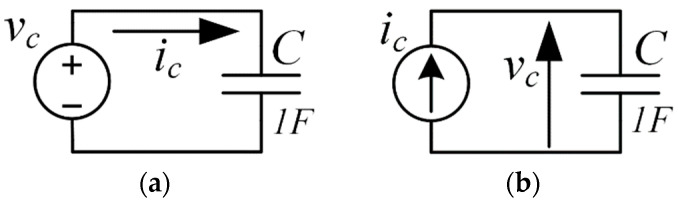
Illustration of Expressions (9) and (10): (**a**) the current $i_{c}$ is the time derivative of the voltage $v_{c}$ and (**b**) the voltage $v_{c}$ is the time integral of the current $i_{c}$.

**Figure 8 sensors-22-05514-f008:**
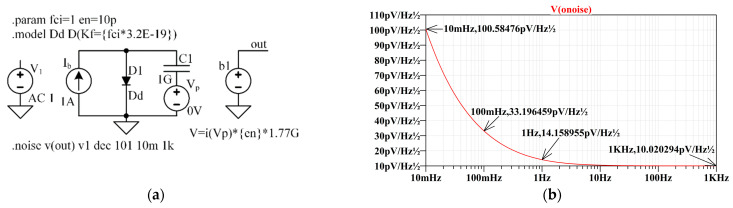
Generation of voltage noise with a desired spectral density $e_{n}$ of the white component and a knee frequency $f_{ci}$ of the flicker noise component. (**a**) The way the circuit looks in the LTSpice.LTSPICE, and (**b**) The LTSPICE graphical output for noise spectral density at the “out” node.

**Figure 9 sensors-22-05514-f009:**
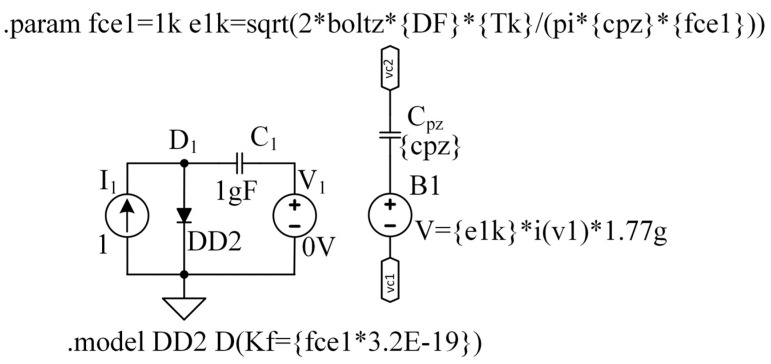
A hierarchical block of the capacitance and the ESR thermal noise of the piezoelectric material as it looks in LTSpice, where vc1 and vc2 are capacitor terminals, cpz is a parameter of the capacitance of the piezoceramic element, boltz is a PSPICE built in Boltzmann’s constant, Tk is a temperature in kelvin, fce1 is the $f_{C_{ESR}}$, and e1k is the $e_{n_{ESR}}$.

**Figure 10 sensors-22-05514-f010:**
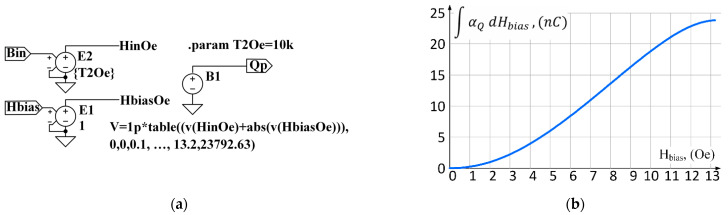
A hierarchical block that has the magnetic signal Bin, in tesla, at its input and electrical charge Qp in coulombs at its output is depicted in (**a**). This is the way the circuit looks in the LTSpice. The DC-magnetic field input Hbias in oersteds sets the bias for magnetostrictive material. The voltage-dependent voltage source E1 has the gain of T2Oe=10k (tesla to oersted factor for free space). The E2 voltage source has a unity gain, and a behavioral voltage source B1 includes a look-up table shown graphically in (**b**) multiplied by the sum of the fields HinOe and HbiasOe.

**Figure 11 sensors-22-05514-f011:**
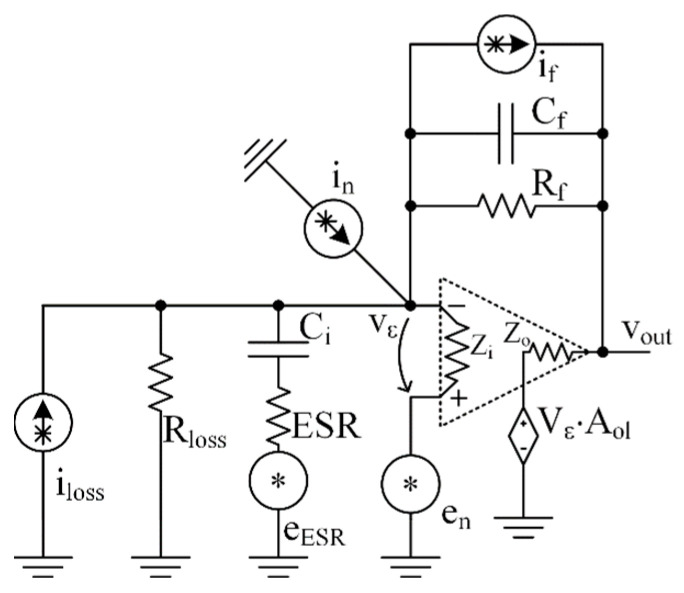
A map of noise sources. The $i_{loss}$ is the current noise corresponding to the thermal noise of the leakage resistance $R_{loss}$ and the $e_{ESR}$ is the ESR voltage noise source. The $e_{n}$ and $i_{n}$ are correspondingly the voltage and the current equivalent input noise sources of the opamp, and $i_{f}$ is the current noise source of the feedback resistor $R_{f}$. All the resistors and opamp are noiseless.

**Figure 12 sensors-22-05514-f012:**
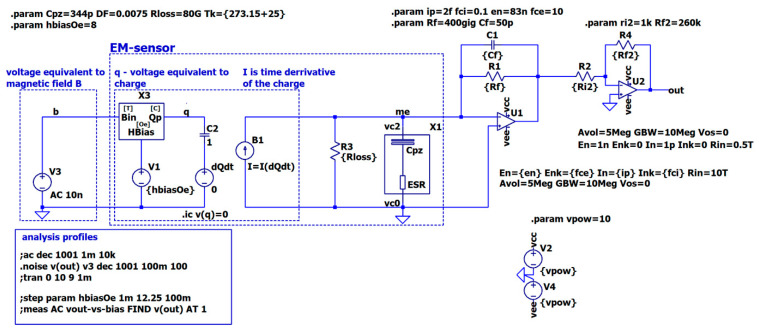
Scheme for simulating an ME sensor with a two-stage charge amplifier implemented in the LTSPICE environment. The hierarchical blocks circuitry X1 and X2 are shown in Figure 10 and Figure 9. Opamps U1 and U2 use a modified built-in “universalopamp” model. All values of each element are provided in the schematic as global parameters under directive **param**. Each circuit element references a specific parameter in curly brackets. All the parameters are described in Table 4.

**Figure 13 sensors-22-05514-f013:**
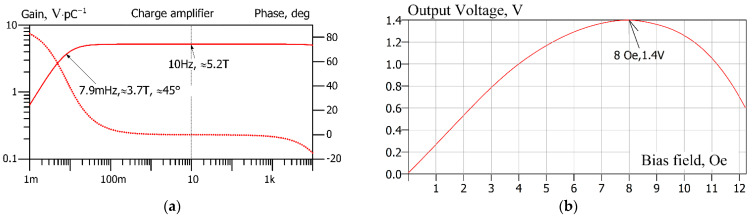
The frequency sweep of the model signals. (**a**) The solid line is a small signal gain (left axis), and the dashed line is the phase (right axis) of the transfer function of the charge amplifier in the frequency range of interest. (**b**) Output voltage at an input signal of 10 nT vs. the bias magnetic field.

**Figure 14 sensors-22-05514-f014:**
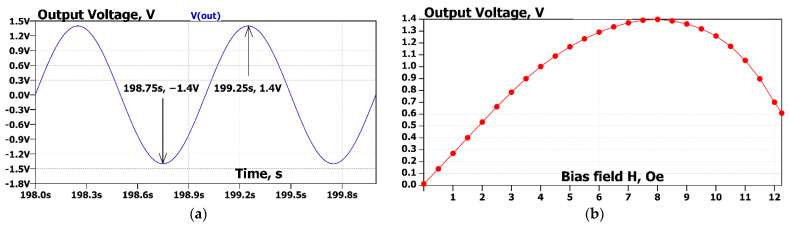
The results of the time-domain (transient analysis) of the system are presented here. The system has a sine wave magnetic field with amplitude $B_{in}=\;10\;\mathrm{nT}$ and frequency f=1 Hz at the input. The simulation time is 200 s. The last two cycles of the output waveform are displayed. (**a**) The output voltage waveform vs. time is shown. The maximum and the minimum values are indicated with arrows. (**b**) The output voltage amplitude is dependent on the bias field. The **step** directive serves for scanning the bias field value (hbiasOe) in the range from 1 mOe to 12.25 Oe with an increment of 500 mOe. The **measure** directive calculates the amplitude for each value of the scanned parameter.

**Figure 15 sensors-22-05514-f015:**
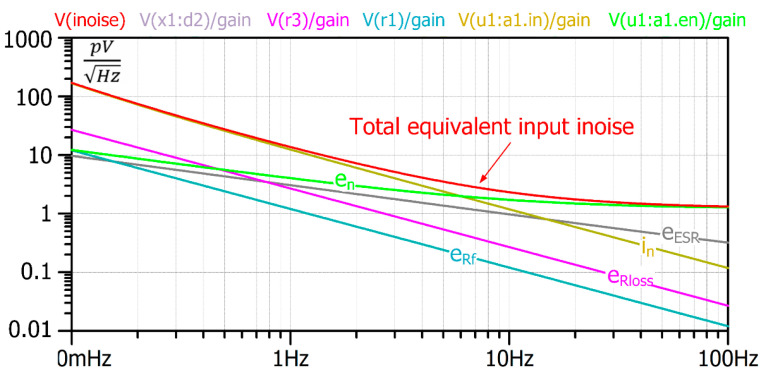
The equivalent input noise and the equivalent input noise components due to each noise source. The dominant noise source $i_{n}$ is the equivalent current noise source of amplifier U1. The equivalent voltage noise units $V/{\mathrm{Hz}}^{1/2}$ correspond to magnetic field noise units $T/{\mathrm{Hz}}^{1/2}$.

**Figure 16 sensors-22-05514-f016:**
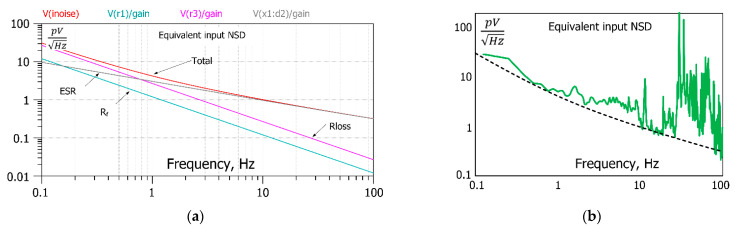
Simulated equivalent input magnetic noise spectral density (**a**) total equivalent input noise spectral density and contribution of dielectric noise, DC resistance noise, and feedback resistor. (**b**) Simulated equivalent input noise spectrum compared with equivalent input noise spectrum measured in [29]. The data is adapted with permission form ref. [29], 2011, John Wiley and Sons.

**Table 1 sensors-22-05514-t001:** Three $G_{x}\left(s\right)$ blocks of the transfer functions corresponding to each of the noise sources.

Noise Sources	$G_{x}\left(s\right)$
$i_{loss},\;i_{n},\;i_{f}$	$G_{lnf}\left(s\right)\approx -R_{loss}∥Z_{C_{pz}}∥R_{f}∥Z_{C_{f}}=-\frac{R_{f}\cdot \;R_{loss}}{R_{f}+R_{loss}+\left(C_{f}+C_{pz}\right)\cdot \;R_{f}\cdot \;R_{loss}\cdot s}$
$e_{ESR}$	$G_{esr}\left(s\right)\approx -\frac{R_{f}∥Z_{C_{f}}∥R_{loss}}{Z_{c_{pz}}+R_{f}∥Z_{C_{f}}∥R_{loss}}=-\frac{C_{pz}\cdot R_{f}\cdot R_{loss}\cdot s}{R_{f}+R_{loss}+\left(C_{f}+C_{pz}\right)\cdot R_{f}\cdot R_{loss}\cdot s}$
$e_{n}$	$G_{n}\left(s\right)\approx 1$

**Table 2 sensors-22-05514-t002:** Parameters of the ME sensor proposed in [29]. Adapted with permission from ref. [29], 2011, John Wiley and Sons.

Parameter of Model	Value
$R_{loss}$	80 GΩ
$C_{PZ}$	344 pF
tan(δ) (or DF)	0.0075
${\alpha_{Q}|}_{H_{bias}\to 8\;\mathrm{Oe}}$	$2680\;\mathrm{pC}\cdot {\mathrm{Oe}}^{-1}$
Charge amplifier gain	$5.18\;V\cdot {\mathrm{pC}}^{-1}$
Charge amplifier BW	0.1Hz<f<100 Hz
Output voltage $V_{out}$, where input signal B=10 nT at f=1 Hz, and bias magnetic field $H_{bias}=8\;\mathrm{Oe}$	1.4 V

**Table 3 sensors-22-05514-t003:** Noise requirements for the operational amplifier and feedback resistor.

Value	0.1 Hz	0.5 Hz	1 Hz	10 Hz	100 Hz
$R_{f},\;G\Omega$	617	123	80	80	80
$i_{n},\;fA/\sqrt{\mathrm{Hz}}$	0.16	0.356	0.516	1.6	5.2
$e_{n},\;nV/\sqrt{\mathrm{Hz}}$	697	328	233	74	24

**Table 4 sensors-22-05514-t004:** Schematic parameters.

Parameter	Description	Value
*ME-sensor:*		
$R_{loss}$	Ceramic dielectric losses equivalent resistance	80 GΩ
$C_{PZT}$	Ceramic equivalent capacitance	344 pF
DF or tan(δ)	Dissipation factor	0.75%
hbiasOe	Biasing magnetic field	8 Oe
*Feedback:*		
$R_{f}$	Feedback resistor	400 GΩ
$C_{f}$	Feedback capacitor	50 pF
*First amp (LMC6044):*		
$A_{vol}$	Open-loop gain	$5\cdot 10^{6}$
GBW	Gain–bandwidth product	10 MHz
$R_{in}$	Input equivalent resistance	10 TΩ
$e_{n}$	Equivalent voltage noise at frequencies >100 Hz	$83\;\mathrm{nV}\cdot {\mathrm{Hz}}^{-\frac{1}{2}}$
$f_{ce}$	Equivalent voltage noise knee frequency	10 Hz
$i_{n}$	Equivalent current noise at high frequencies	$2\;\mathrm{fA}\cdot {\mathrm{Hz}}^{-\frac{1}{2}}$
$f_{ci}$	Equivalent current noise knee frequency	-------
power	Voltage range	±10 V
Second opamp		
$A_{vol}$	Open-loop gain	$5\cdot 10^{6}$
GBW	Gain–bandwidth product	10 MHz
$R_{in}$	Input equivalent resistance	0.5 TΩ
$e_{n}$	Equivalent voltage noise at frequencies >100 Hz	$1\;\mathrm{nV}\cdot {\mathrm{Hz}}^{-\frac{1}{2}}$
$f_{ce}$	Equivalent voltage noise knee frequency	-------
$i_{n}$	Equivalent current noise at high frequencies	$1\;\mathrm{pA}\cdot {\mathrm{Hz}}^{-\frac{1}{2}}$
$f_{ci}$	Equivalent current noise knee frequency	-------
power	Voltage range	±10 V

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
