# Peer review of "Static, Dynamic, and Signal-to-Noise Analysis of a Solid-State Magnetoelectric (Me) Sensor with a Spice-Based Circuit Simulator"

_sensors, 2022, doi:10.3390/s22155514_

Round 1

Reviewer 1 Report

11)    Line 64: the abbreviation «SPICE» must be deciphered.

22)    Lines 172: You need to write how the ME sample is fixed or free; how are the alternating and bias magnetic fields directed; are there electrodes, if there are, where and how they are located, where is measured the electric charge or electric voltage. It is unclear in which quasi-static or resonant mode the ME effect is being investigated. You must give the values of all the necessary material parameters Metglas and PMN-PT piezo-fiber element. You must give all the necessary initial equations, show that the equations describing your equivalent electrical circuit are really equivalent to the original equations. You need to give expressions for ?????, ???, ?? through all the initial values. 

33) Lines 181: typo in the designation of the constant bias field.

Author Response

The authors are very grateful for your kind attention to our work and for essential questions and valuable recommendations to improve the presentation of our study. We appreciate your efforts, and we will try to answer your questions as clearly as possible and make the necessary changes to the body of the article. Please find the answers in the attached document. This document contains answers to questions from reviewers marked RxAy, where x is the number assigned by the editor to the reviewer, and y is the number of the answer to the related reviewer’s question.

Reviewer 2 Report

In this manuscript, the circuit of the magnetoelectric sensor with charge amplifier is simulated based on SPICE simulation software, and the static and dynamic behaviors of the system are qualitatively studied. However, the following issues should be addressed before publication.

(1) When analyzing the circuit simulation of the magnetoelectric sensor with charge amplifier, the specific value of each element of its equivalent circuit should be provided for quantitative analysis, and the specific value of LCR in the circuit is not given.

(2) When analyzing the equivalent model of the magnetoelectric sensor, only the impedance spectrum of the magnetoelectric sensor is not enough to support the model.

(3) There are many illustrations and tables in the article. The illustrations and tables of the full text should be placed in the right place for better reading. Besides, the figures of the whole article should be unified as follows,

A. Format in Figure 10 (b) needs to be corrected.

B. Formats of Figure 13 (a) and (b) are not uniform, and the Y-axis of Figure 13 (a) has no unit.

C. The y-axis unit in Figures 14 and 15 should be corrected with 1/2 in the superscript.

D. Figure 15 (b) has no x-axis and y-axis.

Author Response

(The authors gave the same response as above.)

Reviewer 3 Report

Comments to the Author:

The author demonstrates a simulation study of Magnetoelectric Sensor. Validation of working principles were performed using SPICE modelling. The work is interesting. However, the shortcomes of this work is the lack of experimental evidence. Here are some comments to improve the manuscript.

Q1. Is there any guideline to control/minimize heat dissipation factor while performing SPICE modelling?

Q2. What is the limit of detection of the proposed sensing platform?

Q3. Usually, magnetoelectric response considerably deteriorates due to the poor piezoelectric property, how the proposed design compensate such degradation?

Author Response

(The authors gave the same response as above.)

Round 2

Reviewer 1 Report

Line 161-162: somewhere here it is necessary to write «the geometry of the interdigitated (ID) electrodes is such that the composite is configured in a multi-push-pull modality». And please read the article Journal of Applied Physics 114, 027011 (2013); https://doi.org/10.1063/1.4812221 In this article, the authors managed to calculate a lot in a case very similar to yours.

Author Response

Dear Reviewer. Your comment is constructive. We have included the corresponding fragment in the text (marked in the article with a pink marker). Also essential is the article you recommended for reading. We also placed the reference to this study in the new version of the article in the appropriate place. Thank you.

Reviewer 2 Report

Authors have addressed my all concerns. I recommend to publish it as is.

Author Response

Thank you very much for your work on the review of our study.

Reviewer 3 Report

The author revised the manuscript according to the comments of the reviewer. therefore, I would like to accept the manuscript for publication.

Author Response

(The authors gave the same response as above.)
